# Mitochondria in the Central Nervous System in Health and Disease: The Puzzle of the Therapeutic Potential of Mitochondrial Transplantation

**DOI:** 10.3390/cells13050410

**Published:** 2024-02-27

**Authors:** Kuldeep Tripathi, Dorit Ben-Shachar

**Affiliations:** Laboratory of Psychobiology, Department of Neuroscience, The Ruth and Bruce Rappaport Faculty of Medicine, Technion—Israel Institute of Technology, P.O. Box 9649, Haifa 31096, Israel; kuldeept@campus.technion.ac.il

**Keywords:** mitochondria, CNS, neural development and function, mitochondria and behavior, mitochondrial transplantation, neuropsychiatric disorders

## Abstract

Mitochondria, the energy suppliers of the cells, play a central role in a variety of cellular processes essential for survival or leading to cell death. Consequently, mitochondrial dysfunction is implicated in numerous general and CNS disorders. The clinical manifestations of mitochondrial dysfunction include metabolic disorders, dysfunction of the immune system, tumorigenesis, and neuronal and behavioral abnormalities. In this review, we focus on the mitochondrial role in the CNS, which has unique characteristics and is therefore highly dependent on the mitochondria. First, we review the role of mitochondria in neuronal development, synaptogenesis, plasticity, and behavior as well as their adaptation to the intricate connections between the different cell types in the brain. Then, we review the sparse knowledge of the mechanisms of exogenous mitochondrial uptake and describe attempts to determine their half-life and transplantation long-term effects on neuronal sprouting, cellular proteome, and behavior. We further discuss the potential of mitochondrial transplantation to serve as a tool to study the causal link between mitochondria and neuronal activity and behavior. Next, we describe mitochondrial transplantation’s therapeutic potential in various CNS disorders. Finally, we discuss the basic and reverse—translation challenges of this approach that currently hinder the clinical use of mitochondrial transplantation.

## 1. Introduction

The traditional view of the mitochondria, the double-membrane organelle, was based on their central role in cellular respiration, a process that converts nutrients into adenosine triphosphate (ATP), the primary energy currency of the cell. At present, mitochondria are conceptualized as a cellular hub that maintains cellular homeostasis, regulates apoptosis, calcium signaling, and cellular oxidative stress states [1,2,3]. In addition, the mitochondrial tricarboxylic acid (TCA) cycle provides metabolites that serve as building blocks for a variety of macromolecules including proteins, lipids, carbohydrates, and nucleotides. Mitochondria also control the innate immune response and are essential mediators of epigenetic processes [4] (Figure 1). Serving as a hub, the mitochondria respond to the cellular state and its energy demands, which are transmitted by cytosolic signaling molecules such as Ca^2+^ and protein kinases and by neurotransmitters [5,6,7,8,9,10]. Maintaining overall cellular and mitochondrial homeostasis is strongly dependent on the coordination between mitochondrial and nuclear DNA [11]. The nuclear genes coordinate the synthesis of proteins essential for mitochondrial structure and function and both nuclear and mitochondrial DNA-encoded factors are involved in the regulation of mitochondrial DNA replication and repair [12]. This review focuses on mitochondrial role in the central nervous system (CNS) in health and disease and the challenges of the transplantation of exogenous mitochondria in the brain. The brain is a unique organ by virtue of comprising different cell types (neurons, oligodendrocytes, and glial cells such as astrocytes, microglia, and ependymal cells) and functional areas with intricate interconnections. Moreover, neurons are post-mitotic cells and therefore their protein and mitochondrial turnover rates are lower than those of other tissues. The neuronal cells are flexible and adaptive along life’s path, being highly sensitive to changes in the environment and to insults. Finally, changes in brain neuronal activity and circuitry can have behavioral manifestations. Therefore, the mitochondrial role in the CNS calls for specific consideration. In addition, we review the therapeutic potential of mitochondrial transplantation in the brain and use this manipulation to enlighten the mitochondrial role in neuronal development, activity, synaptic plasticity, and ultimately behavior. The open questions and challenges relevant to mitochondrial transplantation strategies are discussed as well.

## 2. Mitochondrial Homeostasis

Mitochondrial biogenesis, together with fission and fusion, is essential for the maintenance of mitochondrial homeostasis, and they play crucial roles in neuronal development and synaptic function. In fission, mitochondria are divided into smaller organelles, leading to the generation of new mitochondria, aiding in the distribution of energy throughout the neurons, and isolating and removing damaged or dysfunctional mitochondria through autophagy (mitophagy). Fusion, on the other hand, involves the merging of mitochondria, which facilitates the exchange of their components and assists in the efficient distribution of energy [13]. The balance between mitochondrial fission and fusion is regulated by evolutionarily conserved proteins. Mitochondria have two closely apposed boundary membranes. Fusion of adjacent mitochondria involves the fusion of the outer mitochondrial membrane (OMM) and is regulated by the mitofusin 1–2 (MFN1 and MFN2) proteins, and that of the inner mitochondrial membrane (IMM) is regulated by the Optic atrophy 1 (Opa1) protein [14,15]. MFN2 is also involved in ER–mitochondria communication for calcium homeostasis [16], while MFN1 partners with Opa1 [17,18]. The dynamin-related protein 1 (Drp1) protein, cycling between the cytosol and the mitochondrial outer membrane, and the fission 1 protein (Fis1) are involved in mitochondrial fission [15]. Drp1’s fission-promoting activity is controlled by various post-translational modifications in its variable domain, such as phosphorylation, SUMOylation, ubiquitination, and S-nitrosylation [19]. The crucial role of mitochondrial fusion for neuronal function was demonstrated, for example in Purkinje cells (PCs), in which MFN2 deficiency led to impaired respiratory complexes’ activity and inner membrane defects, causing cerebellar dysfunction. *MFN1* and *MFN2* overexpression restored PC viability [20]. Reduction of MFN2 in mitochondria also affects their movement and distribution in axons, leading to abnormal mitochondrial clustering in cell bodies and proximal axons [21]. Opa1 undergoes alternative splicing and proteolytic processing which results in multiple variants that can play different roles in fusion, cristae integrity, and maintaining mitochondrial DNA (mtDNA). Inhibition of Opa1-mediated mitochondrial fusion reduced synaptic marker proteins, decreased synapse numbers, and affected dendritic growth [22,23]. Drp1 is enriched at neuronal terminals and is essential for synapse formation and synaptic sprouting as discussed previously [24]. It was also shown that the inhibition of Drp1 is neuroprotective under toxic conditions such as glutamate toxicity or oxygen–glucose deprivation and ischemic brain damage in neuronal cultures and the brain, respectively [25]. Studies suggest an additional mitochondrial outer membrane protein the ganglioside-induced differentiation-associated protein 1(GDAP1), preferentially expressed in neurons, that, together with Drp1, MFN1–2, and Opa1, is essential for mitochondrial fusion/fission dynamics [26].

Mitochondrial biogenesis is an additional process that regulates mitochondrial homeostasis and is stimulated by cell energy demands and by mitochondrial dysfunction. One key downstream effector of nicotinamide adenine dinucleotide (NAD^+^) is the NAD^+^-dependent histone deacetylase Sirtuin 1 (SIRT1), which regulates the transcription factor peroxisome proliferator-activated receptor-gamma coactivator 1-alpha (PGC-1α), a master regulator of mitochondrial biogenesis. Another PGC-1α activator affected by increased NAD^+^ levels is AMP-activated protein kinase (AMPK), which, in concert with SIRT1, improves metabolic fitness. PGC-1α is an upstream inducer of mitochondrial genes by positively affecting the activity of some hormone nuclear receptors [Peroxisome proliferator-activated receptor gamma (PPARγ) and estrogen-related receptor alpha (ERRα)], which regulate multiple pathways involved in cellular energy metabolism within and outside mitochondria, and of nuclear transcription factors 1 and 2 (NRF-1 and 2). Activation of NRF-1 and 2 leads to the transcription of nuclear-encoded respiratory chain components and mitochondrial transcription factor A (Tfam), thus promoting the synthesis of mitochondrial proteins, mtDNA replication, and transcription (Figure 1). PGC-1α also mediates the crosstalk between fission/fusion processes and mitochondrial biogenesis [20,27]. Although highly important for neuronal homeostasis, development, and survival there is limited knowledge on the mechanism evoking mitochondrial biogenesis in neuron endings away from the cell body. One suggestion is that additional regulatory mechanisms have to be involved for axon-localized mitochondrial biogenesis to occur, such as mechanisms that utilize axonal-localized mRNAs translated into mitochondrial proteins and form contacts between mitochondria and endosome/lysosome platforms [28,29]. Although unlikely, there is evidence that mitochondrial biogenesis can occur in the cell body with the new mitochondria transported by an anterograde mechanism while the damaged mitochondria are moved by retrograde transport and then subjected to mitophagy. Mitochondrial biogenesis is usually assessed by mtDNA content and levels of PGC-1α and its affected transcripts. Variations in mitochondrial biogenesis were shown to be associated with differences in schizophrenia penetrance in a 22q11.2 deletion syndrome [30]. In AD for example, there are reports on enhanced and reduced mitochondrial biogenesis, suggesting that neurons in different stages of degeneration or different brain areas show different patterns of biogenesis [31,32,33,34]. In sum, mitochondrial fission and fusion together with mitochondrial biogenesis affect cellular signaling pathways, gene expression, and cell fate in the complex landscape of neuronal development, synaptogenesis, and astrocyte-neuronal concerted regulation of synaptic plasticity [15,26,35,36].

## 3. Mitochondria and CNS Development

The mature brain, although constituting just 2–3% of the total body weight, consumes nearly 20% of the body’s basal metabolic rate; while early in development, the human brain consumes more than 40% of the body’s basal energy metabolism [37,38,39]. Glucose is the main energy substrate in the brain [40]. Its metabolism, through glycolysis, the non-oxidative breakdown of glucose to pyruvate and lactate in the cytoplasm, and through the tricarboxylic acid (TCA) cycle and oxidative phosphorylation (OxPhos) system in the mitochondria, produces the high-energy molecule ATP. Hence, it is not surprising that mitochondria play an integral role in neuronal development, migration, and synaptic function during CNS development. Neurulation is the series of processes transforming the neural plate into the neural tube, requires organelle biogenesis, including ribosomes and mitochondria, which demands the synthesis of nucleic acid, epigenetic modifications, and changes in precursor cell transcriptomes [41]. During progressive differentiation, both nucleic acid synthesis and epigenetic modifications rely on de novo purine synthesis and raw material generation for methyltransferases. These processes are partially dependent on mitochondrial folate-mediated one-carbon metabolism (FMOCM) (Figure 1). An additional important enzyme involved in the de novo synthesis of pyrimidine nucleotides and is important for neuronal development is dihydroorotate dehydrogenase (DHODH), located in the inner mitochondrial membrane [42,43]. As the neural tube closes, the expression of folate pathway genes is upregulated [44]. It was reported that low levels of dietary folate and mutations in genes encoding FMOCM enzymes are associated with neural tube closure defects [45,46]. Such mutations include those in *Slc25a32*, which encodes for the mitochondrial folate transporter/carrier (MFTC) [47]. Mitochondrial FMOCM, therefore, plays an essential role in ensuring that mitochondria supply the substantial demands for nucleic acids during this early period of neural development. The role of mitochondria in nuclear DNA methylation is not only linked to the folate/methionine cycle activities but also to mtDNA haplotypes. This has been demonstrated by the different degrees of nuclear DNA methylation in tissues and cells containing identical nuclei. According to recent findings, epigenetic histone methylation is also regulated by mtDNA heteroplasmy, [48,49].

At the cellular level, dividing neuronal stem cells (NSCs) in the neuronal tube differentiate into neuronal progenitor cells (NPCs), which either proliferate or further differentiate into neurons or glial cells that constitute the functioning neural network. This process is termed developmental neurogenesis. The determination of NPCs to undergo either proliferation or further differentiation is controlled by both mitochondrial biogenesis and bioenergetics. During the switch from glycolytic metabolism to mitochondrial OxPhos, NSCs become NPCs and subsequently differentiate into various neuronal cell types [50,51,52]. Mitochondrial structure and network dynamics are also crucial for neurogenesis. As NSCs differentiate, the morphology and dynamics of mitochondria undergo significant alterations [53,54]. In the initial state of mouse NSCs, mitochondria appear simple, with a round shape and few cristae. However, during differentiation, they transform into a more complex tubular network consisting of elongated mitochondria with abundant cristae and a dense matrix [55,56,57].

It was further demonstrated that the manipulation of the fission/fusion dynamics of the mitochondrial network, by interfering with the fission and fusion proteins Drp1 and MFN1/2, respectively, impairs neurogenesis and differentiation [58,59]. We have shown that deficits in mitochondrial function and the fusion protein Opa1 were associated with impaired differentiation of dopaminergic and glutamatergic neurons from schizophrenia-derived iPSCs [60]. Neuronal polarity is fundamental for the construction of neuronal networks and circuitry. It was reported that depletion of mitochondria prevented axon formation, while interference with its Ca^2+^ buffering impaired the formation of neuronal polarity [61].

Recently, it was demonstrated that mitochondrial metabolism sets the tempo of neuronal development. To our opinion, this elegant study is a fundamental proof for a mitochondrial role in neurodevelopment. In human and mouse cortical neurons which were xenotransplanted into the mouse brain developed along their species-specific timeline, suggesting that species-specific developmental timing is controlled by cell-intrinsic mechanisms. Furthermore, mitochondria contributed to species-specific developmental rates, as increasing mitochondrial metabolism in cultured human neurons enhanced neuronal maturation, while decreasing mitochondrial metabolism of mouse neurons reduced their maturation rate [62].

Understanding the underlying mechanisms of mitochondrial cross-talk with the cell’s intrinsic mechanism is essentially unknown, yet could be crucial in assessing the involvement of mitochondria in neurodevelopmental and neurodegenerative diseases.

## 4. Mitochondrial Migration in Neurons

Neurons rely on mitochondria for their growth, survival, and functionality; consequently, it is imperative for neurons to exhibit specialized mechanisms facilitating the distribution of mitochondria to distal synapses where energy demand is notably high. The redistribution of mitochondria is subjected to modifications induced by activity-dependent remodeling of both axons and synapses. Mitochondrial transport in neurons is a dynamic and complex process, it involves both anterograde and retrograde movement along microtubules [63]. The anterograde movement depends on a kinesin motor protein, specifically kinesin 1. Kinesins (KIFs) bind to mitochondrial surfaces and use ATP to move them to microtubule ends, which are usually axon terminals. Microtubules are polarized structures that control the direction of transport. The plus ends of microtubules point towards axon terminals, while minus ends are typically located near cell bodies. The mitochondrial retrograde transport from axon terminals to cell bodies is facilitated by a dynein motor protein. Dynein moves along microtubules in the opposite direction of KIFs, utilizing ATP for energy. Thus, the retrograde transport facilitates mitochondrial recycling to maintain their health [31]. Mitochondria attach to the motors through their respective motor adaptor proteins and mitochondrial receptors. In response to increased action potential firing rates or glutamate receptor activation, the complex of KIF5 (member of kinesin-1 family)–Milton (a motor adaptor protein)–MIRO (mitochondria rho family GTPase) mediates the movements of mitochondria along axons and dendrites. Elevated Ca^2+^ concentrations suppresses mitochondrial mobility through a MIRO Ca^2+^-sensing pathway [64,65]. Axonal mitochondria are anchored by syntaphilin (SNPH), a neuron-specific axonal protein that associates with the mitochondrial outer membrane. Studies showed that *SNPH* deletion significantly increased the percentage of mobile axonal mitochondria [31]. Interestingly, a study in Caenorhabditis elegans showed that during regeneration, mitochondria translocate into injured axons to increase the average mitochondrial density upon injury, whereas axons that fail to increase the mitochondrial density exhibit poor regeneration [66]. In mice, enhancing axonal mitochondrial transport by deletion of *SNPH*, improved the regeneration process [67,68]. In developing neurons, SNPH expression is reduced, while during maturation its expression is enhanced, suggesting a high mitochondrial transport demand during development [69]. Armcx1, a mammalian-specific protein that encodes a mitochondria-localized protein, also regulates mitochondrial movement and thereby neuronal health. Hence, its overexpression enhanced mitochondrial transport in axons and the regeneration and neuronal repair of axon outgrowth in retinal ganglionic cells and in cultured cortical neurons, respectively [70]. In dendrites, experimental reduction of mitochondrial number resulted in a loss of synapses and dendritic spines, whereas increasing mitochondrial content or activity increased spine number and synapse plasticity [71]. In all, the ability of mitochondria to move forwards and backwards along axons and dendrites in response to neuronal local-specific demands is essential for neuronal function, development, and repair, as well as for mitochondrial quality control.

## 5. Mitochondria and the Synapse

Mitochondria are involved in synaptic functions through energy and ATP production, as constant release of neurotransmitters and maintenance of ion gradients requires energy. In addition, Ca^2+^, which is firmly regulated by mitochondria within the presynaptic terminal, is a key mediator in neurotransmitter release. The TCA cycle is involved in the metabolism of the main excitatory and inhibitory neurotransmitters, glutamate and GABA, respectively (Figure 1). Notably, mitochondria and the TCA cycle act differently in astrocytes and neurons [72,73]. For example, GABA conversion to glutamine occurs through the astrocytic TCA cycle, since the enzymes glutamine synthetase and pyruvate carboxylase are expressed only in astrocytes, and not in neurons [74]. In neurons, the TCA cycle is involved in converting glucose to glutamate in presynaptic neurons [71,75]. Disorders linked to GABAergic transmissions, such as epilepsy, show compromised synaptic GABA concentrations. Notably, antiepileptic drugs like tiagabine hinder the astrocytic uptake of GABA, leading to an increase in synaptic GABA concentrations [76], suggesting that the astrocyte-specific mitochondrial TCA cycle becomes hyperactive in GABA-impaired disorders. This hyperactivity in mitochondrial energy supply could potentially disrupt the balance of energy distribution between neuronal and astrocytic TCA cycles.

Synaptic activity can generate reactive oxygen species (ROS), which are potentially harmful molecules for the cell. Mitochondria help to buffer and detoxify ROS to protect the delicate structures within the synapse from oxidative damage, while dysfunctional mitochondria produce ROS. We have shown that mitochondrial dysfunction is associated with impaired neuronal sprouting, synaptic connectivity, and neurotransmission in SZ-iPSCs-derived dopaminergic and glutamatergic neurons [60]. In addition, an experimental reduction of mitochondrial number in dendrites resulted in a loss of synapses and dendritic spines, whereas increasing mitochondrial content or activity increased spine number and synapse plasticity [77]. Overall, impairment in mitochondrial function and morphology can influence synaptic transmission and intricate neuron–astrocyte connections, which can lead to either synapse malfunction or neurodegeneration.

## 6. Mitochondria in Behavior

The impact of mitochondrial dysfunction on behavioral impairments has gathered significant attention in the field of neuroscience. As discussed in the previous sections, mitochondrial dysfunction can lead to disruptions in synaptic transmission, impaired neuronal plasticity, and increased susceptibility to neuroinflammation. These factors collectively contribute to the manifestation of various behavioral impairments, including cognitive decline, motor deficits, and mood disorders [12,59].

Cognitive impairment is a common behavioral abnormality in several neurological disorders, such as Alzheimer’s disease (AD), fronto-temporal dementia, and schizophrenia. Ample evidence implicates a role for mitochondrial dysfunction in cognitive impairments, primarily in AD and its progression. A role for synaptic mitochondria and tau proteins in stress-induced cognitive impairment was shown [78]. Thus, long-term stress caused tau-dependent atrophy in dendrites and spines in the prefrontal cortex, associated with alterations in mitochondrial OxPhos, transport, and localization, as well as cognitive impairments. A large-scale proteomic analysis of the human brain identified mitochondrial proteins that were associated with cognitive stability in aging [79]. Numerous studies have linked mitochondrial ROS production, which is mainly induced by the inhibition of the OxPhos system, to age-related and AD cognitive decline, specifically as treatment with mitochondria-targeted antioxidants tends to improve cognition in experimental models and humans (for a reviews see for example [80,81]). In an elegant study using pharmacological means, a causal link was shown between anxiety and social behavior under stress and impaired mitochondrial respiration, ATP deficits, and enhanced ROS production in the nucleus accumbens [82]. In the honeybee, brain mitochondrial function has been found to change with aggressive social interactions [83,84,85]. In humans, age-related increases in aggressive tendency was associated with increased baseline brain mitochondrial respiration [86,87]. Additionally, diet restriction and ketone body feeding, which enhance mammalian brain mitochondrial function, can also increase aggression, suggesting that mitochondrial function plays a role in modulating aggression-related neuronal signaling [84]. Interestingly, mitochondrial dysfunction has been observed in various psychiatric primary behavioral disorders including schizophrenia, bipolar disorder, major depressive disorder, and autism [88,89,90]. Almost all studies show an association between mitochondria and behavior, while the causative link is missing.

## 7. Mitochondria and CNS Diseases

Forming such an important cellular hub and being essential players in cellular survival and death, it is comprehensible that mitochondria are strongly involved in pathological processes leading to numerous general and CNS diseases. Mitochondrial malfunction can be attributed to multiple mitochondrial factors including their enzymes’ function and their mtDNA. Mitochondria in cells are not a homogenous population. A recent study showed, for the first time, two distinct populations of mitochondria in rat liver, cytoplasmic mitochondria and lipid droplet-associated mitochondria. The lipid droplet mitochondria were more efficient in fatty acid oxidation and exhibited specific molecular markers such as carnitine palmitoyl transferase (CPT1), phosphorylated acetyl- CoA carboxylase (ACC), and MFN2, while cytoplasmic mitochondria had higher respiration capacity. Interestingly, mitochondria associated with lipid droplets in a non-alcoholic fatty liver disease (NAFLD) rat model showed compromised fatty acid oxidation, highlighting the role of droplet-associated mitochondria in NAFLD and the significance of functional separation of mitochondria to treat NAFLD [91] and possibly other disorders.

The essential role of mitochondria in neurons has been described above. Any distortion in their activity can lead to changes in synaptic plasticity and neuronal network connectivity, eventually resulting in behavioral changes and neurodegeneration. Astrocytes have a crucial role in supporting neurons, and changes in their mitochondrial function are linked to the maintenance of brain health. Astrocytes typically have low activity of mitochondrial OxPhos as compared to neurons. Nevertheless, it was demonstrated that their OxPhos is essential for degrading fatty acids and maintaining lipid balance in the brain. Astrocytic OxPhos malfunctions can cause accumulation of lipid droplets and elevated acetyl-CoA levels, inducing astrocyte reactivity. The astrocytes’ lipid-centric mechanism was shown to stimulate neuronal fatty acid oxidation and oxidative stress, to activate microglia, and hinder the synthesis of essential lipids for myelin replenishment. As a consequence synaptic loss, demyelination, and neuroinflammation were observed, leading to cognitive decline and neurodegeneration relevant to AD and maybe also to multiple sclerosis (MS) [92]. Ample evidence from genetic, transcriptomic, and proteomic studies supports an association between mitochondrial enzyme deficiencies, specifically those of the OxPhos, and the pathophysiology of CNS disorders, including mental illness, such as schizophrenia, major depression, and bipolar disorders as well as neurodegenerative disorders such as Parkinson’s disease (PD) and AD. For example, in schizophrenia, major depression, and bipolar disorder, a deficiency in the first complex of the OxPhos system, NADH/ubiquinone oxidoreductase (complex I) is repeatedly reported, while in PD and AD, as well as in aging, a deficit is observed in both complex I and the fourth complex, cytochrome-c-oxidase [93,94,95,96,97,98,99,100,101,102].

Mitochondrial TCA cycle intermediates have an essential role in the synthesis of various biomolecules and signaling molecules, which are involved in chromatin modifications, DNA methylation, immunity, and hypoxic response (for a review see [103]). Hence, TCA cycle enzymes have been associated with various CNS dysfunctions. For example, in schizophrenia, enzymes of the first half of the TCA cycle (aconitase, alpha-ketoglutarate dehydrogenase complex, and succinyl coenzyme A synthetase) showed lower activity, while those in the second half (succinate dehydrogenase and malate dehydrogenase) showed higher activities, probably compensating for the lower activity levels of the first half of the cycle [104]. In AD, deficits in brain TCA cycle enzymes, specifically pyruvate dehydrogenase subunit beta (PDHB), succinate-CoA ligase [ADP-Forming] Subunit Beta (SUCLA2), and malate dehydrogenase 1 (MDH1) were observed [105]. Finally, patients with inherited deficiencies in TCA enzymes such as fumarate hydratase (FH) and α-ketoglutarate dehydrogenase (α-KGDH) present, among other symptoms, progressive and severe encephalopathy, psychiatric and pyramidal symptoms, and developmental delay [106,107].

mtDNA also contributes to CNS diseases, mainly due to the strong inflammation associated with increased circulating cell-free (cf)-mtDNA levels (Figure 1). Similar to cf-nuclear DNA [108], the role of cf-mtDNA is increasingly recognized in immune-mediated systemic and central inflammatory diseases [109,110]. The measurement of cf-mtDNA concentration emerges as a potential biomarker for acute inflammatory stress. Damaged mtDNA released into the cytosol activates inflammatory pathways, including the inflammasome and NF-κB, contributing to inflammation at various levels [111,112]. Notably, elevated levels of mtDNA are observed in plasma in CNS disorders with strong inflammatory responses, such as relapsing–remitting multiple sclerosis [113,114]. An increase in cf-mtDNA may reflect early inflammatory activity, leading to mitochondrial damage, neural loss, and brain atrophy and its change has been suggested as a predictor for treatment outcome [115,116]. Interestingly, a reduction in melatonin, an antioxidant produced by neurons and shown to be involved in mitochondrial homeostasis, can lead to disruption of mitochondria, the release of mtDNA, and activation of a cytosolic inflammatory response in neurons, by triggering the cGAS/STING/IRF3 pathway [117]. In PD, it has been suggested that mitochondrial dysfunction can lead to the release of mitochondrial damage-associated molecular patterns (mtDAMPs), triggering neuroinflammation through microglial immune receptors. Conversely, inflammatory molecules released by the glial cells can further damage mitochondrial function, triggering a vicious circle that can lead to neurodegeneration [118,119]. In mental disorders such as major depression and schizophrenia, higher levels of cf-mtDNA were associated with symptoms of depression and schizophrenia. Concomitantly, treatment with certain mood stabilizers (lamotrigine, valproic acid, or lithium) was associated with lower cf-mtDNA [120,121,122,123]. In addition to alterations in cf-mtDNA, changes in the mtDNA genome were also shown to be associated with CNS disorders. For example, SNPs in two control regions in the mtDNA (T16519C and T195C) as well as in mtDNA-encoded cytochrome B (*CYTB*), which protein is a component of the ubiquinol–cytochrome c reductase complex (complex III), were associated with schizophrenia and bipolar disorders. A12308G was associated with psychosis in both disorders [124]. Several point mtDNA mutation deletions and methylation, as well as changes in mtDNA copy numbers, have been associated with neurodegenerative disorders including PD and AD, amyotrophic lateral sclerosis, and Huntington’s disease (for a review see [125]). In monozygotic twins discordant for schizophrenia, a discordance in mtDNA copy number estimates was observed suggesting post-zygotic mtDNA changes that may contribute to the discordance in the disease between monozygotic twins [126]. mtDNA copy number in peripheral blood cells has been suggested as a potential biomarker as it was reduced in PD, increased in attention-deficit hyperactivity disorder (ADHD), and associated with psychosis severity and antipsychotic treatment in schizophrenia [127,128,129]. Taken together, all these findings corroborate that mitochondrial multifaceted dysfunction contributes to the pathogenesis of CNS disorders.

## 8. Mitochondrial Intercellular Transfer and Uptake Mechanism of Transplanted Mitochondria

Mitochondria transfer between cells to reach a homeostatic bioenergetic state in nearby cells or to support the energy demands of injured cells. Studies have shown that intercellular mitochondrial transfer occurs through various contact modes between cells, including gap junctions, cell fusion, and tunneling nanotubes (TNTs) forming between heterogeneous cells. The specific transfer method can vary depending on the cell type involved. TNTs are filled with actin bundles that enable cargo transfer, while microtubules are essential to allow the transfer of organelles as big as mitochondria [130,131,132,133]. Mitochondria, traditionally known as intercellular organelles, were shown to be secreted into the extracellular space, probably via extracellular vesicles (EVs), under both physiological and pathological conditions. These EVs can be taken up by cells and play diverse roles, regulating metabolism, provoking immune response, and inducing cell differentiation in the target cells [134,135]. For example, in mice it was shown that labeled mitochondria transferred from infused bone marrow stem cells to injured lung cells, apparently through the formation of nanotubes and EVs [136,137]. In the CNS, it has been shown that mitochondria can be transferred from neurons to adjacent astrocytes to be degraded or recycled [138]. Astrocytes, on the other hand, were shown to release functional mitochondria that enter nearby neurons by a calcium-dependent mechanism involving CD38 and cyclic-ADP-ribose signaling. These astrocyte-derived mitochondria rescued the nearby neurons in mice subjected to focal cerebral ischemia [139]. Mitochondrial transfer between cells has been studied as a potential therapeutic tool, mostly using mesenchymal stromal cells (MSCs) as donors of healthy mitochondria, in various disease models including respiratory, cardiac, and neurological diseases [68,136,140,141,142,143,144].

An additional route to affect the intercellular mitochondrial state is the transplantation of isolated mitochondria. As early as 1974, Knowles showed that transplantation of isolated mitochondria to Paramecium aurelia can transfer erythromycin-resistance to sensitive cells [145]. In 1982, Clark and Shay showed in mammalian cells that chloramphenicol resistance could be transferred to sensitive cells by mitochondrial transplantation [146]. Thereafter, ample evidence substantiated the ability of exogenous mitochondria to enter cells and improve their bioenergetic state. Several mechanisms have been suggested for the penetration of isolated mitochondria into cells including caveolae-dependent clathrin-dependent endocytosis, actin-meditated endocytosis, and micropinocytosis [147,148,149,150,151]. However, in contrast to intracellular transfer and export/uptake of mitochondria containing EVs, the mechanism by which isolated extracellular mitochondria enter the cell is still obscure. Regardless, transplanted exogenous mitochondria can fuse with the recipient cell’s endogenous mitochondrial network, which in cells results in enhanced cellular respiration and ATP content [152,153]. Attempts have been carried out to increase the penetration efficiency of the exogenous mitochondria. For example, it was shown that mitochondria packaged in cell membrane lipid rafts were taken up by cultured cells twice as much as unpackaged mitochondria. In another study, labeling mitochondria with Pep-1, an amphipathic peptide carrier and a member of the cell-penetrating peptide family, significantly increased the efficiency of mitochondrial penetration in pheochromocytoma (PC12) cells and mouse brains treated with 6-hydroxydopamine (6-OHDA), a PD-inducing agent, and in a mitochondrial disease model of myoclonic epilepsy with ragged red fibers. In both models, mitochondrial function was improved [152,154]. The question remains whether artificially increasing mitochondrial penetration is beneficial in all cell types and physiological states, as cellular control on uptake is bypassed.

## 9. Mitochondrial Transplantation in CNS Disorder Models

During the past decade, studies examining the therapeutic potential of transplantation of isolated extracellular mitochondria in a variety of general and CNS disease models have thrived (for reviews see [134,155,156,157] and Table 1). The beneficial effects of exogenous mitochondrial transplantation have been demonstrated in diverse pathological models including cardiac, lung, liver, and CNS pathologies [158,159,160,161,162,163,164,165]. In humans, a few clinical trials showed beneficial effects of autologous mitochondrial transplantation, but the follow-up period was relatively short. In infants who required ECMO support for ischemia–reperfusion-associated myocardial dysfunction, autologous mitochondrial transplantation led to an improvement in ventricular function and release from ECMO support. However, there is no information on how stable the improvement was [166]. Another human study was performed on six children with single large-scale mitochondrial DNA (mtDNA) de novo deletion syndromes (SLSMDs), a rare and severe multisystemic disease. Mitochondrial augmentation therapy, in which the maternal mitochondria, mostly similar to those of the recipient, were transplanted into the patients’ enriched hematopoietic cells following leukapheresis, was employed. Following the transplantation procedure, partial and limited improvement in mitochondrial number and clinical symptoms was observed during 6–12 months of follow-up [167]. In women with recurrent pregnancy failures, transplantation of autologous mitochondria to mature human oocytes with sperm at the time of intracytoplasmic sperm injection resulted in a significant improvement in the ratio of good-quality embryos and healthy normal babies [168]. All of these studies used autologous or allogeneic maternal-derived mitochondria for transplantation rather than allogeneic mitochondria to reduce immune response and avoid heteroplasmy.

Several studies have transplanted exogenous mitochondria in various CNS experimental models. In 6-OHDA Parkinson’s disease rat model, transplantation of exogenous allogeneic and xenogeneic mitochondria coupled with Pep-1 into the medial forebrain bundle (MFB) reduced loss of dopaminergic neurons in the substantia nigra three months later, suggesting that the effect of the transplanted mitochondria spread beyond the injection site, probably due to the characteristics of the MFB, being a neural pathway containing fibers. The restoration of the dopaminergic neurons was associated with enhanced mitochondrial functions, reduced neuroinflammation, and enhanced locomotive activity [155]. In this study, no significant difference was observed between Pep-1-coupled xenogeneic and allogeneic mitochondria-induced effects. The same group showed similar beneficial bioenergetical and behavioral effects at three to four weeks following repeated chronic (once a week for three months) intranasal injection of Pep-1-labeled mitochondria [165]. Another study used the MPTP null mouse model of PD and intravenously injected them with exogenous mitochondria. The GFP-labeled transplanted mitochondria were distributed in various organs including the liver, kidney, muscle, and brain. MPTP mice systemically injected with mitochondria showed improved locomotion, reduced ROS generation, and restored ATP levels and complex I activity. Systemically injected mitochondria in healthy mice did not affect ATP levels nor spontaneous locomotion but significantly increased latency to immobility in the forced swimming test [169]. Both studies showed improvements in cell survival and mitochondrial functions in cell cultures treated with the relevant toxins. Injection of exogenous mitochondria into the tail vein was also performed in a mouse model of AD, produced by intracerebroventricular injection with amyloid-beta 1–42 peptide. One to two weeks after the intravenous injections of fresh human-isolated mitochondria, AD mice exhibited significantly improved cognitive performance. Furthermore, there was a notable reduction in neuronal loss and gliosis in the hippocampus and increased activities of citrate synthase and cytochrome c oxidase in treated AD mice compared to non-treated AD mice [162]. Acute effects of intravenous repeated transplantation of mitochondria isolated from young mice into aged mice included improved mitochondrial bioenergetics and reduced redox state, associated with ameliorated learning and motor functions in the aged mice [170]. The protective effect of exogenous mitochondrial transplantation was also studied in experimental spinal cord injury (SCI). One study reported that up to 7 days post-transplantation, mitochondria were observed in multiple resident cell types but not in neurons. The other study detected the transplanted mitochondria up to 28 days post-transplantation in the vicinity of the lesion. One day after transplantation, a partial restoration of mitochondrial respiration as well as amelioration of mitochondrial fragmentation and cellular apoptosis were observed. Partial functional protection, assessed by tissue paring and recovery of sensory and motor function, was observed four weeks after transplantation; however, it faded after 6 weeks of follow-up [171,172]. In traumatic brain injury, which caused mitochondrial impairments, anxiety and cognitive deficits, transplantation of allogeneic liver-isolated mitochondria restored astrocytic brain-derived neurotrophic factor (BDNF) levels and the behavioral deficits [173].

The past decade has witnessed an abundance of studies focusing on mitochondrial abnormalities in several mental disorders including major depression and schizophrenia. A wide array of methodologies ranging from imaging through genetic, biochemical, and molecular to histological and structural techniques were used to reveal multifaceted mitochondrial dysfunction in mental disorders, specifically in schizophrenia [88,174,175]. Hence, mitochondrial transplantation effects were also assessed in experimental models of mental disorders. In a lipopolysaccharide-induced mouse model of inflammation-induced depression, intravenous injection of exogenous mitochondria acutely reduced depressive-like behaviors assessed by forced swim, tail suspension, and sucrose preference tests. This was associated with an acute reduction in astrocyte and microglia activation and cytokines levels, higher levels of BDNF transcripts and neurogenesis, and restored mitochondrial dysfunction measured by ATP and ROS production in mouse hippocampi [176]. We have studied the effect of exogenous allogeneic mitochondrial transplantation in schizophrenia patient-derived iPSCs (SZ-iPSCs) and in the maternal immune activation Poly I:C rat model of schizophrenia. Transplantation of mitochondria into SZ-iPSCs restored mitochondrial deficits including mitochondria respiration, ΔΨ_m,_ mitochondrial network dynamics, and transcript levels of specific subunits of complex I and of OPA1. Concomitantly, an enhanced efficiency of SZ-iPSCs differentiating into functional dopaminergic and glutamatergic neurons was observed. In the rat model of SZ, intra-medial prefrontal cortex (mPFC) single injection of mitochondria, in adolescent rats (34 days old), restored mitochondrial impairments and neuronal outgrowth and activity. assessed by monoamines’ transmission. in adulthood (100–120 days old). Proteomics analysis of the mPFC showed an association between the beneficial neuronal and mitochondrial effects and improved metabolic and neuronal development and plasticity pathways. Finally, mitochondrial transplantation in Poly I:C rats restored schizophrenia-related behavioral deficits such as attentional deficit and spontaneous locomotor activity in a novel environment. The behavioral changes showed a significant correlation with changes in monoamine and neuronal structural alterations. Unlike the intra-MFB injection in the PD model, our data suggest a localized effect of the transplanted mitochondria. Unexpectedly, a similar injection protocol to healthy rats induced detrimental effects in all parameters mentioned above, including behavioral, bioenergetical, and neuronal-related features. These data emphasize the importance of the bioenergetical and physiological states of the recipient in the outcome of mitochondrial transplantation. Furthermore, the opposite effects induced by mitochondrial transplantation in the schizophrenia model and healthy rats advocate for a causal link between the mitochondria and behavior, neuronal activity and plasticity [163,164].
cells-13-00410-t001_Table 1Table 1Research reports on mitochondrial transplantation in CNS disease models.DiseaseSpecies/Applied ModelSource of MitochondriaRoute of TransplantationRef.Parkinson’s diseaseRatAllogeneic/xenogeneic Pep-1-labeled mitochondriaInjection into the medial forebrain bundle[154]Parkinson’s diseaseMouseHuman hepatoma cells Intravenous injection[169]Parkinson’s diseaseRatLiver allogeneic mitochondria conjugated with Pep-1Intranasal infusion [165]Alzheimer’s diseaseMouseHeLa cell-derived mitochondriaTail intravenous injection[162,177]Cognitively impaired aged mice MouseLiver allogeneic mitochondria isolated from young miceInjected into the hippocampus of aged mice [178]Diabetes-associated cognitively impaired miceMousePlatelet-derived mitochondriaIntracerebroventricular injection[179]Chronic mild stressed—aged ratsRatBrain-derived mitochondria from young ratsInjected intracerebroventricularly in aged rats[180]AgingMouseLiver-derived allogeneic mitochondria from young ratsIntravenous injection[170]CNS injuryMouseCerebral cortex-derived allogeneic mitochondria whose DJ1 protein was modified with O-GlcNAcylation Intraventricular injection [181]Traumatic brain injuryMouseLiver/muscle-derived autologous mitochondriaInjected into the cerebral cortex[173]Brain strokeMouseBone marrow mesenchymal stem cell-derived allogenic mitochondriaIntranasal administration[182]SchizophreniaRatRat brain-derived allogeneic mitochondriaBilateral injection of into medial prefrontal cortex[163,164]Lipopolysaccharide (LPS)-induced model of depressionMouseHippocampus-derived allogeneic mitochondriaIntravenous injection[176]Spinal cord injuryRatSoleus muscle-derived allogeneic mitochondriaInjected into spinal cord[171]Spinal cord ischemiaRatSoleus muscle-derived allogeneic mitochondria Intravenous transplantation via the jugular vein[183]Spinal cord injuryRatSoleus muscle-derived allogenic mitochondriaInjection into spinal cord via intraparenchymal route[172]Neuromuscular limb injuryMouseNRSystemic injection[184]Lower limb ischemia–reperfusion injuryMouseHuman umbilical cord mesenchymal stem cell-derived mitochondriaGastrocnemius muscle injection[185]Optic nerve injuryRatLiver-derived allogeneic mitochondriaIntravitreal injections[186]Mitochondria transplantation in humans Recurrent pregnancy failure casesHumanOvarian cortical tissue-derived autologous mitochondria Mitochondria transferred to human oocytes, intracytoplasmic sperm injection[168]Ischemia–reperfusion-associated myocardial dysfunctionHuman (infants)Rectus abdominis muscle-derived autologous mitochondriaInjected into the myocardium affected by ischemia–reperfusion[166]Single large-scale mitochondrial DNA (mtDNA) de novo deletion syndromes (SLSMDs)Human (infants)Maternal PBMC-derived mitochondria transplanted into the patients’ enriched CD34+ cells after leukapheresisTransfused into patients[167]NR—not reported.

## 10. Challenges of Mitochondrial Transplantation

The evidence described above supports the potential of mitochondrial transplantation to become a strategy for reducing behavioral and cellular injuries associated with mitochondrial dysfunction in general and CNS disorders. However, there are still concerns that need to be addressed both at the level of basic mechanisms and at the level of reverse-translational or bedside-to-benchtop research, which are summarized in Figure 2.

The first relates to the ability of mitochondria to penetrate into organs’ cells and to cross the blood–brain barrier. As discussed earlier, several uptake mechanisms have been proposed, although none has clearly described the molecular mechanism by which mitochondria are taken up or how they escape the endosomes or macrophagosomes. Another related question is how mitochondria retain their integrity. The answer to this question is still an enigma. The integrity of the exogenous mitochondria is of special concern in blood, where Ca^2+^ concentrations are around 1.8 mM as opposed to 100 and 200 nM in the cytosol. Such high Ca^2+^ concentrations are known to induce the opening of permeability transition pores and thereby blunting mitochondrial membrane potential eventually leading to mitochondrial swelling and destruction [187]. In line with the integrity of the transplanted mitochondria is the question of their half-life survival in cells.

There have been attempts to follow the exogenous mitochondria both in cell cultures and in vivo. Different means for follow-up have been reported, including labeled mitochondrial proteins, their labeling with a cationic mitotracker that indicates the maintenance of mitochondrial membrane potential, and donor mtDNA, which differs from that of the recipient. As discussed previously, data on the survival rates of exogenous mitochondria range from a few days up to 12 weeks. This diversity is probably linked to the means used to follow the mitochondria, as mtDNA survival differs from that of mitochondrial proteins, which also show a high diversity in their half-life rates and are tissue-dependent. It is accepted that half-lives of intact mitochondria, their DNA, and proteins are in the range of days, ~8–28, 20–30, and ~7–30 days, respectively, depending on the tissue and the specific labeled protein used for follow-up. An additional issue in following mitochondrial integrity is the existence of EVs containing free mtDNA and mitochondrial proteins [135,188,189,190], suggesting that following mitochondrial integrity is challenging. Given thisthe above mentioned time scale, it is conceivable that transplanted exogenous mitochondria will stay intact for no more than several days. mtDNA may last longer than integral mitochondria, yet it is conceivable that the foreign mtDNA will be eliminated by the host cell upon mismatch, as the nDNA–mtDNA coordination is essential to cell survival [191]. 

Still, mitochondrial transplantation exerted long-term bioenergetical and behavioral effects. An attractive hypothesis is that the transplanted mitochondria induce local bioenergetics or immune responses, being strongly immunogenic [192]. A plausible complementary hypothesis is that the exogenous mitochondria or their content are transferred to adjacent cells from the recipient cells either via TNTs or by secreted EVs, both changing the physiological homeostasis in the recipient cells, which eventually lead to long-term effects. In line with the latter are the findings that intravenously injected mitochondria, observed in the liver but not in the brain, improved cognitive functions and our findings of a crosstalk between the physiological state of the recipient and exogenous mitochondria determining the long-term beneficial/detrimental outcomes [162,164].

The therapeutic potential of mitochondrial transplantation has also been questioned [193,194,195]. One major problem is the duration of the beneficial effect and whether treatment can be repeated in humans. Another concern is the source of the transplanted mitochondria, as in many mitochondrial-related disorders the malfunction of mitochondria is observed throughout the body, and therefore allogeneic mitochondria will be necessary for transplantation. Such mitochondria can increase the susceptibility to provoking inflammation, oxidative stress, and the risks of heteroplasmy [181,196]. In addition, it brings up the ethical problems of introducing a foreign DNA. A foreign mitochondrial genome may interfere with the nuclear genome, causing mitochondrial genetic drift and thereby mitochondrial dysfunction [197]. Opinions are widely divergent about the extent of this risk, with some arguing that human studies will almost certainly show adverse effects [198], while others argue there will be little risk [199]. Certainly, the lack of clarity on this matter poses a risk and is a subject of further research, which is currently underway [200]. The above-described concerns are augmented in systemic administration, which necessitates a considerably higher dose of mitochondria and a much longer delivery time to reach a target cell, and as many different organs receive the mitochondria it is less specific [201].

Finally, mitochondrial transplantation can induce both beneficial and detrimental bioenergetical, neuron structural, and behavioral alterations depending on the recipients’ mitochondrial homeostasis as well as on their physiological state, of which the specific relevant factors are currently an enigma. Therefore, any future use of mitochondrial transplantation for therapeutic aims will have to consider the possible toxic effects induced by the two-edge sword mitochondria.

## 11. Conclusions

Mitochondria are integral to cellular processes, particularly within the CNS, which emphasizes their vital importance in maintaining overall brain function. The multifaceted functions of CNS mitochondria, discussed in this review, range from their involvement in neuronal development, synaptogenesis, and plasticity to their adaptation to the complex interactions between brain cells. Mitochondrial transplantation is an innovative approach for the treatment of CNS disorders. However, the application of translational research to clinical implementation needs to be carefully considered in the light of the concerns discussed above. Nevertheless, mitochondrial transplantation provides an essential tool to study the causal link between mitochondrial function and neuronal development, sprouting, and plasticity, as well as behavior in health and disease. Given the essential role of mitochondria in numerous pathologies, interference with mitochondrial activity is of utmost importance and thus requires future intensive research.

## Figures and Tables

**Figure 1 cells-13-00410-f001:**
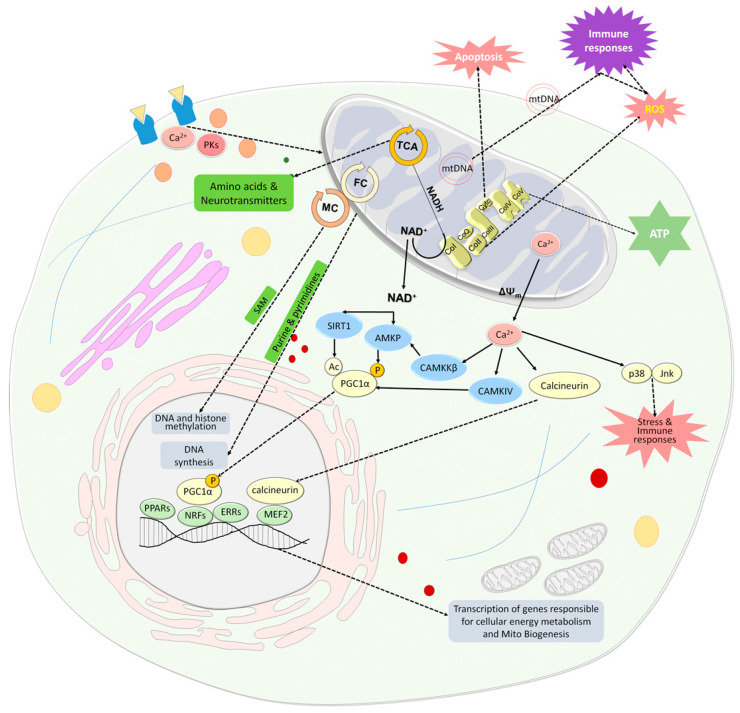
Mitochondria are an important cellular hub. Mitochondria produce energy in the form of ATP through the OxPhos system, which is also a major source of ROS and is involved in apoptosis. The TCA cycle not only provides reduced substrates in the form of NADH and NADPH for the respiratory chain, but also provides amino acids, that are converted to neurotransmitters and serve as building blocks for proteins. The folate cycle, which is located in the mitochondria and the cytosol, provides purines and pyrimidines for DNA synthesis and, together with the methionine cycle, provides methyl groups for DNA and histone methylation. In addition, mitochondria communicate with the nucleus in homeostasis and stress. Changes in the NAD^+^/NADH ratio and extracellular Ca^2+^ concentrations are two major inducers of signaling pathways that activate PGC-1α. An increase in NAD^+^ levels is associated with reduced ATP levels and increased extracellular Ca^2+^ concentration is linked to dissipation of ΔΨm and enhanced leak of mitochondrial Ca^2+^. NAD^+^ activates SIRT1 which in turn activates PGC-1α by its deacetylation. NAD^+^ also activates AMKP which in turn activates PGC-1α by its phosphorylation. AMPK can also be activated by CaMKKβ, which is activated by Ca^2+^. PGC-1α is an upstream inducer of mitochondrial genes. It positively affects the activity of some hormone nuclear receptors, PPARs and ERRs, which regulate multiple pathways involved in cellular energy metabolism within and outside mitochondria, and nuclear transcription factors, NRFs, which leads to the transcription of nuclear coded respiratory chain components and Tfam. PGC1α can be also activated by CaMKIV which in turn is activated by Ca^2+^. Ca^2+^ also activates the phosphatase calcineurin, which activates the nuclear MEF2, which in turn regulates many muscle-specific genes and lipogenic and glycogenic enzyme genes. Ca^2+^ released from the mitochondria induces a stress state via the activation of stress-related MAPKs, p38 and Jnk, which leads to the activation of the immune system and oxidative stress state as do cell-free mtDNAs. Mitochondria are also involved in hemoglobin and steroid production and respond to the cellular state and its energy demands, transmitted by cytosolic signaling molecules such as Ca^2+^ and protein kinases and by neurotransmitters. Abbreviations: Ac—acetyl group; AMPK—AMP-activated protein kinase; CaMKIV—Ca^2+^/calmodulin-dependent protein kinase type IV; CaMKKβ—Ca^2+^/calmodulin-dependent protein kinase kinase-β (CaMKKβ); Co—complex; ERRs—estrogen-related receptors; FC—folate cycle; Jnk—c-Jun NH2-terminal kinase; MC—methionine cycle; MEF2—myocyte-specific enhancer factor 2; mtDNA- mitochondrial DNA; NAD—nicotinamide adenine dinucleotide; NFRs—nuclear respiratory factors; OxPhos-oxidative phosphorylation; P—phosphate group; p38—p38 mitogen-activated protein kinase; PGC-1α—PPARγ co-activator 1α; PK—protein kinases; PPARs—peroxisome proliferator-activated receptors; ROS—reactive oxygen species; SAM—S-adenosylmethionine; TCA—tricarboxylic acid; Tfam—mitochondria transcription factor A; ΔΨm—mitochondrial membrane potential.

**Figure 2 cells-13-00410-f002:**
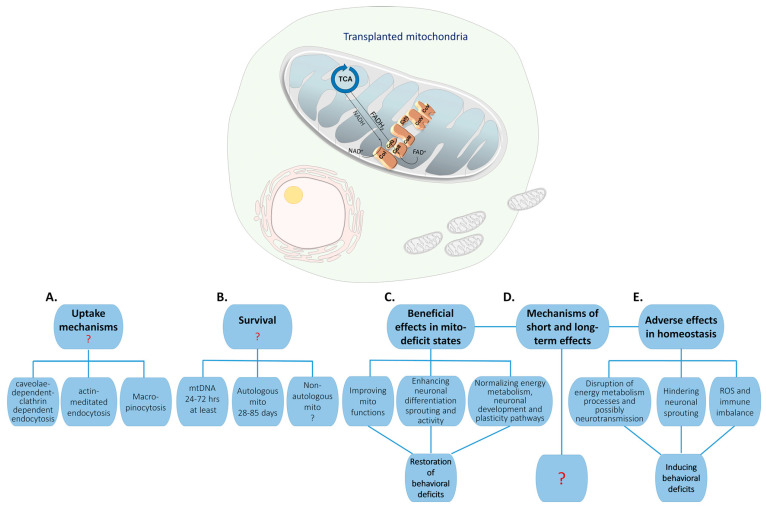
Challenges in understanding the fate and the mechanism of action of transplanted mitochondria in the CNS: Transplanted mitochondria (mito), whether autologous or allogeneic, are taken up by cells. Several uptake mechanisms have been suggested (**A**), but none has been systematically investigated. The survival rate of the transplanted mitochondria (**B**) is still obscure as it depends on many factors including cell type, the methodology used to measure survival, mitochondrial proteins or mtDNA, and the source of the mitochondria, i.e., whether they are autologous or non-autologous. Regardless, it is accepted that the survival of the transplanted mitochondria has a shorter half-life than their induced long-term morphological, molecular, and behavioral effects have (**C**,**D**). Notably, it was shown that the changes induced by the transplanted mitochondria into the CNS or iPSCs depend on the endogenous mitochondrial bioenergetics state and probably on the physiological state of the recipient ((**C**) vs. (**D**)). The mechanisms controlling the long-term changes, which ultimately lead to behavioral alterations, are still an enigma (**E**).

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
