# Peer review of "Mitochondria in the Central Nervous System in Health and Disease: The Puzzle of the Therapeutic Potential of Mitochondrial Transplantation"

_cells, 2024, doi:10.3390/cells13050410_

Round 1
Reviewer 1 Report
Comments and Suggestions for Authors
Present manuscript provides the review of current knowledge about the mitochondrial role in the CNS. In addition, authors are focused on mitochondrial transplantation as tool to study the causal link between mitochondria and neuronal activity and behaviour as well as its therapeutic potential in various CNS disorders. Finally, they discuss the basic and reverse-translation challenges of this approach that currently hinder the clinical use of mitochondrial transplantation. Manuscript is well written by the persons that are real experts in the field. However, there are some minor points that should be modified.
In addition to the cf-mtDNA, mtDNA level in leukocytes can be associated with some disorders of CNS and could reflect the relative content of mitochondria in the affected peripheral tissue. For example, reduced mtDNA copy number was detected in substantia nigra pars compacta but not in other parts of brain and it was also reflected by decreased mtDNA copy number in leukocytes of the patients with Parkinson's disease (Pyle et al., 2016). On contrary, higher mtDNA copy number in leukocytes was documented in association with ADHD (Kim et al. 2019). Decreased mtDNA copy number in the leukocytes of schizophrenia patients was also documented (Fizikova and Racay, 2022) and mtDNA copy number in leukocytes of schizophrenia patients could be associated with psychosis severity and anti-psychotic treatment (Kumar et al. 2018).
Kim JI, Lee SY, Park M, Kim SY, Kim JW, Kim SA, et al (2019) Peripheral Mitochondrial DNA Copy Number is Increased in Korean Attention-Deficit Hyperactivity Disorder Patients. Front Psychiatry. 10: 506.
Kumar P, Efstathopoulos P, Millischer V, Olsson E, Wei YB, Brüstle O, et al (2018) Mitochondrial DNA copy number is associated with psychosis severity and anti-psychotic treatment. Sci Rep. 8: 12743.
Pyle A, Anugrha H, Kurzawa-Akanbi M, Yarnall A, Burn D, Hudson G (2016) Reduced mitochondrial DNA copy number is a biomarker of Parkinson's disease. Neurobiol Aging. 38: 216.e7-216.e10.
Fizikova I, Racay P (2022) Oxidative modifications of plasma proteins and decreased leukocyte mitochondrial DNA of schizophrenia patients. Act. Nerv. Super. Rediviva. 64: 25–32.
There are some typo errors. E.g.
Mitochondrial Migration in neurons
Page 7 line 324 87]. In
Page 8 line 376 in schizophrenia
Author Response
Reviewer 1:
We thank the reviewer for encouraging review and the important comments, which contribute to the merit of the MS. We revised the manuscript according to the comments and corrected typos and style. We hope that in its revised form the MS will be acceptable to the reviewer.
In addition to the cf-mtDNA, mtDNA level in leukocytes can be associated with some disorders of CNS and could reflect the relative content of mitochondria in the affected peripheral tissue. For example, reduced mtDNA copy number was detected in substantia nigra pars compacta but not in other parts of brain and it was also reflected by decreased mtDNA copy number in leukocytes of the patients with Parkinson's disease (Pyle et al., 2016). On contrary, higher mtDNA copy number in leukocytes was documented in association with ADHD (Kim et al. 2019). Decreased mtDNA copy number in the leukocytes of schizophrenia patients was also documented (Fizikova and Racay, 2022) and mtDNA copy number in leukocytes of schizophrenia patients could be associated with psychosis severity and anti-psychotic treatment (Kumar et al. 2018).
The reviewer raised important information regarding mtDNA copy number in blood cells. We have added this information in line 419: “mtDNA copy number in peripheral blood cells has been suggested as a potential biomarker as it was reduced in PD, while increased in attention-deficit hyperactivity disorder (ADHD) and associated with psychosis severity and antipsychotic treatment in schizophrenia (Pyle et al., 2016, doi: 10.1016/j.neurobiolaging.2015.10.033; ÖÄŸütlü et al., 2020, doi: 10.24869/psyd.2020.168; Kumar et al., 2018, doi: 10.1038/s41598-018-31122-0)”.
Reviewer 2 Report
Comments and Suggestions for Authors
The review “Mitochondria in the CNS in Heath and Disease; the Puzzling of the Therapeutic Potential of Mitochondrial Transplantation” by Kuldeep Tripathi and Dorit Ben-Shachar overviews some aspects of the role of mitochondria (mt) in health and disease (indeed, reviewing all of them would be a mammoth task) with particular focus on the transplantation of mt, as a therapeutic tool. Thus, authors briefly describe the overall role of mt, mt fission/fusion, role of mt in CNS development, mt migration/transport in neurons, mt in synaptic functions, behavioral impairments, some CNS diseases.
Please find below my comments and suggestions.
All abbreviations (CNS, ROS, etc) should be defined upon first use
Note regarding all figures – font is too small, text is hard to read (its almost illegible). Could you please increase the size of the font and make text darker/more visible?
Line 46. “mitochondrial health” - what does it mean? Perhaps you should re-write this part of the sentence using “scientific writing style”
Line 54. “Moreover, neurons are post-mitotic cells and therefore proteins as well as mitochondrial turnover rates are lower in brain than in peripheral
tissues”
Neurons are not the only cells in the brain (and such cells other than neurons are not post-mitotic). Could you please briefly list all (predominant) types of cells in the brain?
Line 65 “through the OhPos system”
The commonly used abbreviation for oxidative phosphorylation is OxPhos – please always use this abbreviation in the text
Line 118. “to mutations in dominant proteins” - what dominant proteins?
Line 137. “One key downstream effector” - downstream of what? please specify
Line 184. please remove the hyperlink (“dihydroorotate dehydrogenase”)
Line 259. “axons that fail to increase mitochondria” - number of mt?
Line 365. Please remove the hyperlink (“in schizophrenia”)
Line 387. An abbreviation “mtDNA” is introduced for the first time. However, the authors use such abbreviation in the text from the very beginning.
Line 602. “As discussed previously, data on the survival rates of exogenous mitochondria
range from a few days up to 12 weeks” Here you can compare it with average half-life of mt in the cell (see line 609)
Line 604. “mDNA” – please fix the typo
Line 607. “This suggests that following mitochondrial integrity is much more complex”. - perhaps you should rewrite this sentence to make it clearer
Line 610 - “it is conceivable that transplanted exogenous mitochondria will stay intact for no more than several days”
What about mtDNA? For how long will transplanted mtDNA stay in the target cell, given the possibility of the fusion of transplanted and endogenous mt?
Are there works in which the authors monitored “half-line” of mtDNA of transplanted mt?
Line- 614 “or immune responses” Are there any works in which non-functional mt were transplanted, to test such hypothesis (regarding mt being trigger of immune response and their beneficial role being caused solely by such response?
Line 617. “the recipient cells in EVs, which can then transfer their cargo to adjacent cells. In line with the latter are the findings that mitochondria injected” - injected as a part of EV's cargo?
What about transport of mt via tunelling nanotubes (TNTs)?
Line 630. “a foreigner DNA” - foreign DNA perhaps?
Line 630. A foreign mitochondrial genome may interfere with the nuclear genome, causing mitochondrial genetic drift [187]” - why it's an issue?
What about mt transplantation in a context of CNS trauma?
https://pubmed.ncbi.nlm.nih.gov/33798765/
https://www.ncbi.nlm.nih.gov/pmc/articles/PMC6980730/
https://www.frontiersin.org/journals/neuroscience/articles/10.3389/fnins.2022.800883/full
etc
Table summarizing recent/seminal works on mt transplantation as a therapeutic tool would be nice addition to the text.
The authors also omit the role of mt in regeneration after trauma (and role of mt in response to trauma in CNS). It might be of interest to discuss (briefly) the role of mt in neuroregeneration (post-trauma), in cell reprogramming in a context of CNS trauma therapy, cell etc.
What about the role of mt in age-associated diseases of CNS other than AD and in healthy ageing of CNS?
Overall, I find that in parts the review just “scratches the surface” of the subject and in some aspects lacks the depth of the narrative. Apart from this (and comments above), it's a very well-written review.
Reviewer 3 Report
Comments and Suggestions for Authors
This is a comprehensive review of a topic of great interest, both for researchers and clinicians. I don't want to make this review longer than it already is, but can the authors write a brief section about the technical aspects of preparing mitochondria for transplantation in vivo ?
Comments on the Quality of English LanguageThe review is very well written. I suggest to correct the few remaining grammatical errors / typos.
Author Response
Reviewer No. 3:
We thank the reviewer for the encouraging review. We corrected typos and style and hope that in its revised form the MS will be acceptable to the reviewer.
Can the authors write a brief section about the technical aspects of preparing mitochondria for transplantation in vivo?
The reviewer suggested adding a paragraph describing the technical aspects of preparing mitochondria for transplantation in vivo. The technical aspects of mitochondria preparation depend on the source of the mitochondria, which differs between studies and is mostly described in their reports. Therefore, we think that adding these technical aspects will add substantially to the length of the MS. We have mentioned a few techniques that were used to improve mitochondrial uptake following transplantation such as lipid rafts (line- 465), Pep-1 (line 466) and mitochondrial augmented therapy (line 485).